# Perfusion Index Analysis in Patients Presenting to the Emergency Department Due to Synthetic Cannabinoid Use

**DOI:** 10.3390/medicina55120752

**Published:** 2019-11-20

**Authors:** Selman Yeniocak

**Affiliations:** Department of Emergency Medicine, University of Health Sciences, Haseki Training and Research Hospital, Istanbul 34130, Turkey; selmanyeniocakacil@hotmail.com

**Keywords:** emergency department, monitoring, organ damage, perfusion index, peripheral perfusion, pulse oximeter, synthetic cannabinoid

## Abstract

*Background and Objectives:* The perfusion index (PI) indicates the ratio of pulsatile blood flow in peripheral tissue to non-pulsatile blood flow. This study was performed to examine the blood perfusion status of tissues and organs of patients using synthetic cannabinoids (SCs). *Materials and Methods:* The records of patients aged 17 or over presenting to the adult emergency department due to SC use between 1 January 2016 and 31 December 2017 were examined in this single-center, retrospective, cross-sectional study. Examined factors included time from consumption of SC to presentation to the emergency department, as well as simultaneously determined systolic and diastolic blood pressures, heart rate (beats per min), Glasgow Coma Score (GCS), and PI values. Patients were divided into two groups, A and B, depending on the amount of time that had elapsed between SC consumption and presentation to the emergency department, and statistical data were compared. *Results:* The mean PI value in Group A was lower than that in Group B. Therefore, we concluded that peripheral tissue and organ blood perfusion is lower in the first 2 h following SC consumption than after 2 h. Systolic, diastolic, and mean arterial blood pressure and mean GCS values were also statistically significantly lower in Group A than in Group B. *Conclusions:* A decreased PI value may be an early sign of reduced-perfusion organ damage. PI is a practical and useful parameter in the early diagnosis of impaired organ perfusion and in monitoring tissue hypoxia leading to organ failure.

## 1. Introduction

Patients presenting as a result of synthetic cannabinoid (SC) use represent an important emergency department (ED) patient group. The use of SCs has increased continually in recent years. This growing use, particularly by young people, has made SCs a health problem and threat to society [1]. SCs exhibit their effects through cannabinoid receptor (CB) type 1 (CB1) and CB type 2 (CB2) [2]. Through these receptors, SCs produce N and P/Q-type calcium channel inhibition by lowering membrane potential with sodium–potassium pump inhibition by interacting with cell membrane voltage-gated ion channels [3]. SCs also exhibit a dose-dependent biphasic effect on the autonomic nervous system. At low and moderate doses, they increase sympathetic activity, reduce parasympathetic activity, lead to tachycardia, and produce an increase in cardiac output. In contrast, at high doses, they inhibit sympathetic activity and increase parasympathetic activity, resulting in bradycardia and hypotension [4].

The perfusion index (PI) represents the ratio of pulsatile blood flow in peripheral tissue to non-pulsatile blood flow and permits the continual measurement of peripheral perfusion in a non-invasive manner using pulse oximetry [5]. As a non-invasive technique, PI values are generally investigated using a pulse oximeter attached to the fingertip, which is generally well perfused with oxygenated blood, or to the ear lobe [6].

Patients generally present to hospital EDs in the acute period following SC use. Therefore, emergency physicians must have a good knowledge of the effects of SCs. Our review of the literature revealed no previous studies investigating tissue and organ blood perfusion due to the potential cardiovascular (CV) effects of SC consumption. The purpose of this study was to investigate the blood perfusion of tissues and organs of patients using SCs and to monitor tissue and organ perfusion thanks to PI monitoring of the clinical status of these patients in the ED.

## 2. Materials and Methods

This research was performed as a single-center, retrospective, cross-sectional study. Following receipt of approval from the Health Sciences University Haseki Education and Research Hospital Specialization in Medical Education Committee under protocol No. 17 dated 17 January 2019, information for patients presenting to the adult ED due to SC use between 1 January 2016 and 31 December 2017 was investigated from the hospital automated information processing system. Then, patient records were retrieved from the hospital archive in the light of that information.

The record files were examined of patients presenting to the ED by ambulance or on foot, assessed by an emergency physician and suspected to have used SCs and treated and follow up accordingly. Information recorded from the patient’s file records included their age, sex, history of disease, drug-use data, and time between SC intake and presentation to the ED, simultaneously determined systolic and diastolic blood pressure (BP), heart rate (beats/min), Glasgow Coma Score (GCS), and perfusion index (PI) data, which was measured with a probe from the second finger using a Massimo Root pulse oximeter capable of non-invasive measurement between 0.02 and 20. Exclusion criteria included patients for whom SC could not be definitely confirmed at evaluation in the ED, whom we concluded had used alcohol or other sedative narcotics in addition to SC, with any CV disease at histories taken from the individual or relatives, with a history of drug use capable of affecting the CV system, or who had deficient data in their records.

Patients included in the study were divided into two groups depending on median values for time to presentation to the ED after SC use. Group A: The patient group presenting to the ED in the first 2 h after SC use. Group B: The patient group presenting to the ED more than 2 h after SC use.

SPSS 15.0 for Windows software was used for statistical analysis. Descriptive statistics were presented as the number and percentage for categorical variables and mean, standard deviation, minimum, and maximum for numerical variables. Since the variables did not meet normal distribution conditions, independent two-group comparisons were performed using the Mann–Whitney U test. Chi-square analysis was used to evaluate categorical variable rates between the groups. Determining factors were examined using linear regression analysis. Cutoff values were determined using ROC (Receiver Operating Characteristic) curve analysis. *p* < 0.05 was regarded as statistical alpha significance.

## 3. Results

We retrieved the records and examined the files of 240 patients presenting to the adult ED due to SC use between 1 January 2016 and 31 December 2017. Following our assessments, we excluded a total of 103 patients, including 24 whose SC use was uncertain, 32 whom we concluded had used alcohol or other sedative narcotics in addition to SCs, one patient due to the presence of CV disease at history taken from the subject or relatives, one patient with a history of use of drugs affecting the CV system, and 45 due to deficient data in their records (Figure 1).

One hundred thirty-seven patients were eventually enrolled, 130 (94.9%) men and seven (5.1%) women, with a mean age of 27.3 ± 9.1 years (range 17–68). Mean time to presentation to the ED following SC use was 3.1 ± 3 h (range 1–24). Patients’ mean systolic blood pressure was 114.1 ± 15.9 mmHg (range 80–195), mean diastolic blood pressure was 70.4 ± 10.8 mmHg (range 40–115), and mean arterial blood pressure was 84.9 ± 11.8 mmHg (range 53.3–135). Mean heart rate was 81.9 ± 18.8 bpm (range 42–136). Mean GCS was 13.8 ± 2.1 (range 4–15), but two patients were not included in that value since they were intubated. Mean PI was 3.16 ± 3.26 (range 0.19–14).

Group A consisted of 78 (56.9%) patients and Group B consisted of 59 (43.1%) patients. Group A patients’ mean systolic, diastolic, and arterial blood pressure, GCS, and PI values were significantly lower than those of the patients in Group B (*p* = 0.009, *p* = 0.004, *p* = 0.004, *p* = 0.001, and *p* < 0.001, respectively). A comparison of various demographic data, mean vital signs, and mean PI values of the subgroups included in the study is summarized in Table 1.

Patients’ mean PI values were significantly positively correlated with time between SC use and presentation to the ED and diastolic blood pressure (*p* < 0.001, and *p* = 0.028, respectively) (Table 2, Figure 2).

Time between SC use and presentation to the emergency department was identified as the single most significant factor in the model established to examine PI levels (*p* < 0.001) (Table 3).

Group B members’ PI cutoff value with 81.4% sensitivity and 83.3% specificity was 1.99 (Positive Predictive Value (PPV): 78.7%, Negative Predictive Value (NPV): 85.5%, Accuracy: 82.5%) (Figure 3).

## 4. Discussion

The mean PI value in Group A was significantly lower than that in Group B. Therefore, we concluded that peripheral organ and tissue perfusion was lower in the first 2 h after SC intake than after the first 2 h. In addition, mean systolic, diastolic and arterial blood pressure, and GCS values were also statistically significantly lower in Group A than in Group B.

PI values are obtained through pulse oximetry measurement of the ratio of the pulsatile signal during arterial flow to the non-pulsatile signal, both values being obtained from absorbed infrared (940 nm) light [5]. Our review of the literature revealed no previous studies investigating the PI in SC-user patients, and ours is the first study on the subject. Under conditions of low blood pressure and circulatory failure, as a result of the vasomotor autoregulation that ensures the continuity of blood flow, the direction of blood flow changes from tissues of less vital importance, such as the skin, subcutaneous tissue, muscle, and the gastrointestinal tract to essential organs such as the brain, heart, and kidney, and the blood flow to peripheral tissue decreases [7]. However, when the imbalance between oxygen (O_2_) requirements and supply is prolonged, vasomotor autoregulation mechanisms become insufficient, and the foundation is laid for multi-organ failure syndrome [8]. Decreased perfusion in cutaneous tissues with no vasomotor autoregulation mechanism may be an early indication of organ damage [7]. Early identification of impaired organ perfusion occupies an important place in the prevention of tissue hypoxia, resulting in organ failure [9]. This study is important in terms of revealing the relation between time to presentation to the ED following SC use and tissue hypoxia that may develop in these patients.

Lima et al. compared capillary filling time, PI, and arterial oxygen saturation in patient and control groups and reported that hypoperfusion in critical patients was correlated with a PI value of 1.4 or less [10]. In their study of septic patients, He et al. reported normal perfusion at a PI value of 1.4 or more, mild perfusion impairment between 1.4 and 0.6, and severe perfusion compromise at 0.6 or less [11]. Another study reported that a PI value of 1.24 or less could be used to estimate the severity of disease in newborns [12]. In the present study, the mean PI value of patients presenting to the ED at any time following SC use was within the normal perfusion range. The PI values in our study in patients presenting to the ED within the first 2 h after SC use were slightly above the reported hypoperfusion value. However, the rise in the mean PI value in our patients in association with time to presentation to the ED creates the suspicion that PI in the first minutes after SC use is lower than the mean PI value determined, and may also be below the critical value. A low PI value associated with SC use occurs through SCs causing CB1 receptor-mediated vasodilation, sympathetic activity inhibition at high doses in the autonomous nervous system, and hypotension developing in association with increased parasympathetic activity. The low diastolic blood pressure observed with SC use in this study may be explained by systolic, diastolic, and mean arterial blood pressure having the same effects of SCs in patients presenting to the ED in the first 2 h after SC use [4,13].

Some studies have reported that heart rates per minute in patients using SCs may be within normal physiological ranges, although both tachycardia and bradycardia have also been reported [4,14]. In our study, heart rates per minute were within normal physiological limits both in patients presenting to the ED in the first 2 h and in those presenting after 2 h, and the difference between the two was statistically insignificant. We think that this arose as a result of the effects of intoxication developing in association with the active agents in the SC combination, the ratio of those agents, the degree of purity, and the quantity used [15,16].

SC use is reported to be more widespread at young ages [1,17]. Hoyte et al. reported a median age of 22.5 years, and that 74.3% of use was among males [17]. Aksel et al. reported a median age of 22 years, and that 96.4% of use was among males [14], while Karabulut et al. reported a mean age of 25.4 years, with 98.6% of use among males [18]. The mean age in our study was 27.3 years, ranging from 17 to 63, suggesting growing SC use at more advanced ages in addition to the young population. The gender distribution of our patients was 94.9% male and 5.1% female. One previous study investigated 852 college students and reported a mean age among SC users of 20.6 years, but that 47% of users were women [19].

Another one of our findings is that two patients were brought to the ED with orotracheal intubation and were admitted to the intensive care unit (ICU). No admission to the ICU was required for any patient other than these two subjects. Apart from the death of one of the patients admitted to the ICU, all patients were discharged in a healthy condition. Monte et al. reported that the majority of their 76 SC users were treated in the ED, with only seven being admitted to the ICU [20]. Küçük et al. determined that 54.1% of patients presenting to the ED due to SC use were discharged from the ED, while 24.1% required intensive care. They attributed their low rate of discharge from the ED to the high rate of admission to the ICU and to physicians’ low level of knowledge concerning SCs [21]. Lank et al. also reported low levels of knowledge concerning SCs among ED physicians [22]. We attribute the lower rates of admission to the ICU and mortality in the present study compared to previous research to the continually changing content of the SCs entering the market, the high rate of SC consumption in our region, to ED physicians gradually becoming very familiar with the spectrum of effects that may occur following SC use, and to all our patients being assessed, monitored, and treated by emergency medicine specialists.

The mean GCS of our patients was less than 15. We attribute this to SC’s CB1 receptor-mediated neurological effects in the brain [23]. SCs have an elimination half-life of 30 h, and are expelled in urine, and partly in bile, by being broken down into inactive metabolites in the liver [24]. The mean GCS value was lower in Group A than in Group B. This may be due to elimination in the period following the first 2 h after SC use, and thus to a weakening of their CB1 receptor-mediated effects [24]. The two patients brought to the ED who were already intubated were not included in the mean GCS value. Of these two patients, one presenting 4 h after SC use had a PI value of 0.91, while the other, a 47-year-old man brought to the ED after 3 h and dying on the same day, had a PI value of 0.27. We attributed this to the SCs entering the market at different times and places having a broad dose-dependent side-effects profile and a continually changing content [25].

### Limitations

Our study was performed in a single center with a limited number of patients. SCs or their metabolites were not subjected to laboratory analysis in any cases. This study is retrospective in nature. Data for all patients presenting to the ED within the specified dates were scanned, and patients meeting the inclusion criteria were included. Therefore, it was not possible to calculate the sample size. We think that more powerful findings might have been obtained had our study been prospective, if PI values had also been determined at time of discharge, and had a control group been established for comparison of mean PI values.

## 5. Conclusions

A decreased PI value may be an early indication of decreased perfusion organ damage. PI may be a practical and useful parameter in the early detection of impaired organ perfusion and in monitoring tissue hypoxia, which is a cause of organ failure.

## Figures and Tables

**Figure 1 medicina-55-00752-f001:**
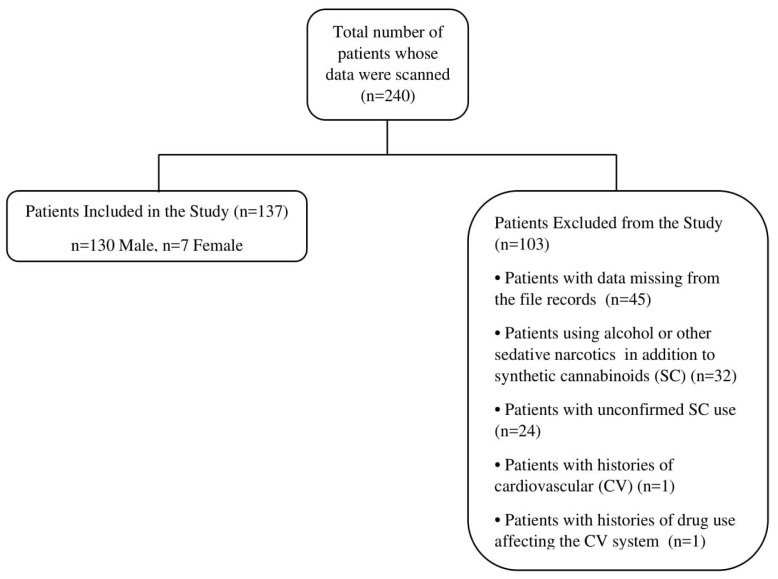
Patient flow chart.

**Figure 2 medicina-55-00752-f002:**
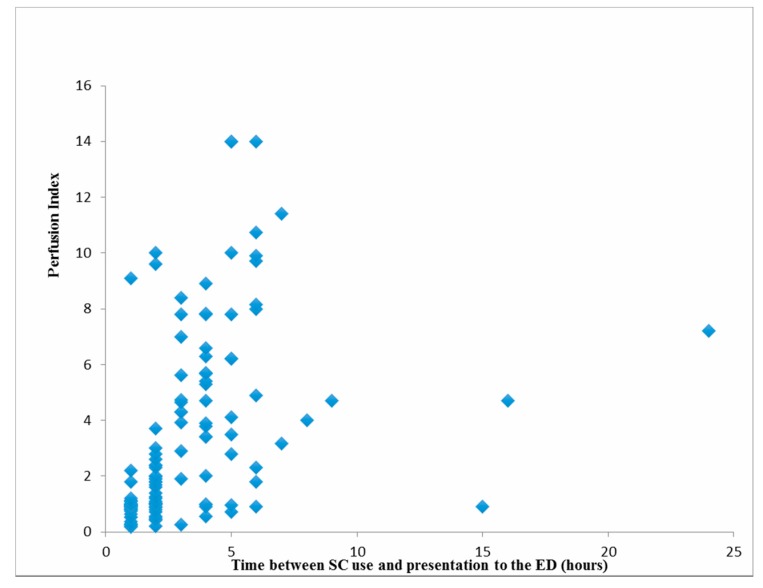
Relations between patients’ mean perfusion index (PI) values and time between synthetic cannabinoids (SC) use and presentation to the emergency department.

**Figure 3 medicina-55-00752-f003:**
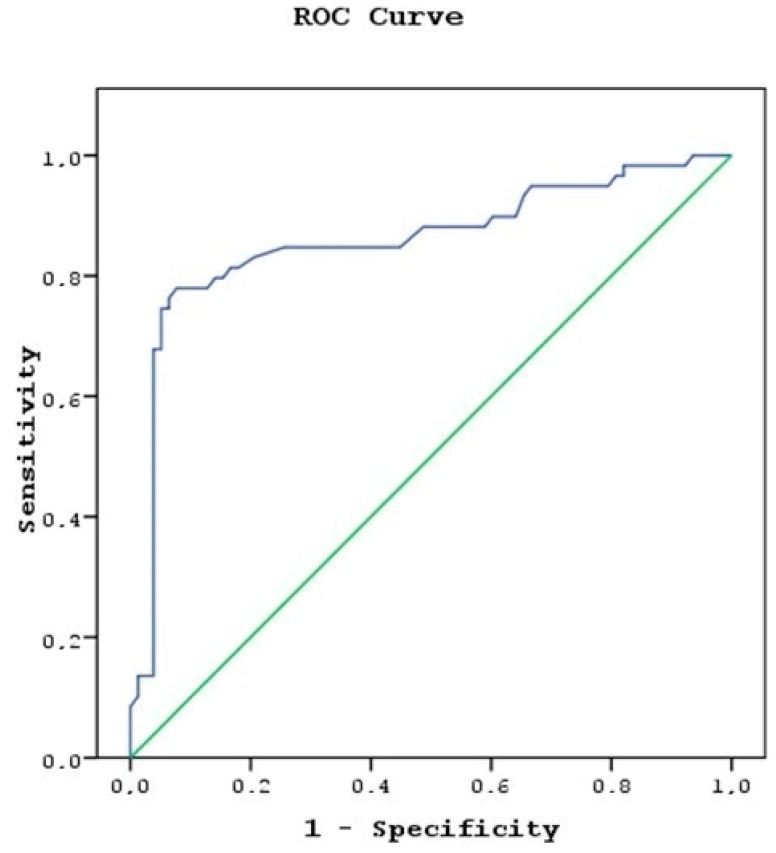
Group B members’ PI cutoff value area under the curve (AUC): 0.858 (95% CI 0.788–0.929).

**Table 1 medicina-55-00752-t001:** Comparison of patients’ and group members’ demographic data, mean vital signs, perfusion index values, and. time between SC use and presentation to the emergency department.

	Total		Group A		Group B	
	Mean ± SD (Min–Max)		Mean ± SD	Median	Mean ± SD	Med
Age	27.3 ± 9.1 (17–68)		27.4 ± 8.9	26	27.2 ± 9.4	2
Gender *n* (%)	Male	130 (94.9)	74 (94.9)		56 (94.9)	
	Female	7 (5.1)	4 (5.1)		3 (5.1)	
Hours elapsed since SC use (h)	3.1 ± 3 (1–24)		1.4 ± 0.5	1.13	5.3 ± 3.4	4
Systolic BP * (mmHg)	114.1 ± 15.9 (80–195)		111.7 ± 16.7	110	117.3 ± 14.2	12
Diastolic BP * (mmHg)	70.4 ± 10.8 (40–115)		68.3 ± 10.7	70	73.1 ± 10.5	7
Mean arterial BP *	84.9 ± 11.8 (53.3–135)		82.7 ± 12.0	80	87.8 ± 10.9	9
Heart rate (beats/min)	81.9 ± 18.8 (42–136)		82.5 ± 18.0	78	81.2 ± 20.0	7
GCS **	13.8 ± 2.1 (4–15)		13.3 ± 2.4	14	14.3 ± 1.4	1
Perfusion Index (PI)	3.16 ± 3.26 (0.19–14)		1.49 ± 1.77	0.99	5.37 ± 3.47	4
Intubation *n* (%)	2 (3.4)		0 (0)		2 (3.4)	

***** BP: Blood Pressure. ****** Two intubated patients were not included in the mean value.

**Table 2 medicina-55-00752-t002:** Patients’ mean PI values and statistical relations. GCS: Glasgow Coma Score.

Perfusion Index		
	rho	*p*-Value
Age	−0.013	0.878
Systolic BP * (mmHg)	0.125	0.146
Diastolic BP * (mmHg)	0.187	0.028
Mean arterial BP * (mmHg)	0.164	0.056
Heart rate (beats/min)	−0.091	0.290
GCS ** Time between SC use and presentation to the ED (hours)	0.130 0.639	0.134 <0.001

* BP: Blood Pressure. ** Two intubated patients were not included in the mean value.

**Table 3 medicina-55-00752-t003:** Linear regression analysis performed to examine factors determining PI levels.

**Enter Method**		**B**	**Beta**	***p*-Value**
	Fixed	2.223		
	Age	−0.018	−0.051	0.533
Hours since intake		0.443	0.408	<0.001
Systolic BP * (mmHg)		−0.019	−0.094	0.448
Diastolic BP * (mmHg)		0.037	0.124	0.324
Heart rate (beats/min)		−0.007	−0.041	0.615
	GCS **	0.116	0.074	0.360
	Sex	−1.232	−0.084	0.301
**Backward Method**	Fixed	1.775		
Hours since intake		0.464	0.428	<0.001

* BP: Blood Pressure. ** Two intubated patients were not included in the mean value.

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
