# Peer review of "Perfusion Index Analysis in Patients Presenting to the Emergency Department Due to Synthetic Cannabinoid Use"

_medicina, 2019, doi:10.3390/medicina55120752_

Round 1

Reviewer 1 Report

Dear authors,

Thanks for submitting your work to the journal. You describe a retrospective analysis showing that the perfusion index (PI) is associated with the time elapsed from the last use of synthetic cannabinoids (SCs).

The strength of the study is its originality. The weaknesses are the fact that no causality can be interpreted from these retrospective analysis. The limitations section should integrate this (the risk of confounders is very high, e.g. the fact that these patients are not coming in the same psychological context when soon vs. late after the use of SCs).

The separation of the patients into two groups is another issue because based on the choice of a threshold. Should be better argued, and ideally the analysis should be more based on the analysis of the continuous variable (see hereafter).

In conclusion, any causal inference must be omitted, as any interpretation regarding 'organ perfusion' or other clinical interpretation.

Minor comments:

It would be interesting to have the regression equation from the linear regression analysis.

It would be useful to check your work using e.g. the STROBE guidelines. Even if not designed for, a patients flow chart can be, for example, easier for the reader. 

Author Response

Dear editor,

I would like thank you and the reviewers for the valuable comments and constructive criticisms concerning manuscript. I have carefully evaluated each comment and corrected the text accordingly. My response to the reviewers’ comments is given below. Thank you for giving us a second chance to resubmit my manuscript.

Yours faithfully

Response to Reviewer 1 Comments

    Reviewer: 1

Comments to the Author

Point 1: It would be interesting to have the regression equation from the linear regression analysis.

Response 1: I agree with the reviewer. Unfortunately, such an equation could not be produced from the existing linear regression analysis.

Point 2: It would be useful to check your work using e.g. the STROBE guidelines. Even if not designed for, a patients flow chart can be, for example, easier for the reader. 

Response 2: I have produced a patient flow chart as recommended by the reviewer. This has been added to the manuscript and is highlighted in yellow.

Note: I have also checked the references and have highlighted various amendments in yellow.

Reviewer 2 Report

Dear Authors 

For proffesionals of emergency medicine inadequate perfusion of organs is a matter of concern influencing mortality rate so I went through the manuscript with interest. 

line 57 - full name of Bioethical Comittee and a number of the consent would be appropriate here. Clinical Trial Registy Number should also be included if the study was registered. 

line 84-86 

I would opt for a graph presenting a number of patients included into the study and excluded from the final anaylsis. 

line 193

I personally do not understand why the authors including 240 patients present such number as a limitation, I would personally opt for sample size calculation supporting the number of patients included into the analysis. 

The study is neither revolutionary nor especially inventive, but woth publicising. In its present form it seems ready for publication after abovementioned minor changes are introduced. 

Best wishes to authors 

Author Response

Dear editor,

I would like thank you and the reviewers for the valuable comments and constructive criticisms concerning manuscript. I have carefully evaluated each comment and corrected the text accordingly. My response to the reviewers’ comments is given below. Thank you for giving us a second chance to resubmit my manuscript.

Yours faithfully

Response to Reviewer 2 Comments

     Reviewer: 2

Comments to the Author

Point 1: line 57: Full name of Bioethical Comittee and a number of the consent would be appropriate here. Clinical Trial Registy Number should also be included if the study was registered. 

Response 1: I have added the reference to the Health Sciences University Haseki Education and Research Hospital Specialization in Medical Education Committee under protocol No. 17 dated 17.01.2019, and this has been highlighted in yellow. I am also forwarding the approval document to the journal.

Point 2: line 84-86: I would opt for a graph presenting a number of patients included into the study and excluded from the final anaylsis. Çalışmaya dahil edilen ve son anaylizden dışlanan bazı hastaları sunan bir grafiÄŸi tercih ederim.

Response 2: I have produced a patient flow chart in the light of both reviewers’ advice. This has been added to the masnucript and highlighted in yellow .

Point 3: line 193: I personally do not understand why the authors including 240 patients present such number as a limitation, I would personally opt for sample size calculation supporting the number of patients included into the analysis. 

Response 3: I agree with the reviewer. This study is retrospective in nature. And data for all patients presnting to the ED within the specified dates were scanned, and patients meeting the inclusion criteria were included. It was not therefore possible to calculate the sample size. This has been added to the limitations section and is highlighted in yellow.

Note: I have also checked the references and have highlighted various amendments in yellow .

Round 2

Reviewer 1 Report

Dear authors, 

You should pay more attention to the reviewer's remarks, that you mostly did not address. 

"The weaknesses are the fact that no causality can be interpreted from these retrospective analysis. The limitations section should integrate this (the risk of confounders is very high, e.g. the fact that these patients are not coming in the same psychological context when soon vs. late after the use of SCs)."

"The separation of the patients into two groups is another issue because based on the choice of a threshold. Should be better argued, and ideally the analysis should be more based on the analysis of the continuous variable (see hereafter)."

"In conclusion, any causal inference must be omitted, as any interpretation regarding 'organ perfusion' or other clinical interpretation."

"The equation of the linear regression should be given ".

Author Response

Dear editor,

We would like thank you and the reviwers for the valuable comments and constructive criticisms concerning our manuscript. We have carefully evaluated each comment and corrected the text accordingly. Our response to the reviwers' comments is given below. Thank you for giving us a second change to resubmit our manuscript.

Yours faithfully
